# Tracing the Genetic Evolution of Canine Parvovirus Type 2 (CPV-2) in Thailand

**DOI:** 10.3390/pathogens11121460

**Published:** 2022-12-02

**Authors:** Tippawan Jantafong, Sakchai Ruenphet, Harold R. Garner, Krit Ritthipichai

**Affiliations:** 1Department of Preclinical Sciences, Faculty of Veterinary Medicine, Mahanakorn University of Technology, Bangkok 10530, Thailand; 2Department of Biomedical Affairs, Edward Via College of Osteopathic Medicine, Spartanburg, SC 29303, USA

**Keywords:** CPV-2, Thailand, viral evolution

## Abstract

Canine parvovirus type 2 (CPV-2) is responsible for hemorrhagic gastroenteritis in dogs worldwide. High genomic substitution rates in CPV-2 contribute to the progressive emergence of novel variants with increased ability to evade the host immune response. Three studies have analyzed the genomic mutations of CPV-2 variants in Thailand. These investigations were independently conducted at different timepoints. Thus, a retrospective integrated analysis of CPV-2 genomic mutations has not been fully performed. Our study aimed at evaluating the evolutionary changes in CPV-2 in Thailand from 2003 to 2019. Two hundred and sixty-eight Thai CPV-2 nucleotide sequences were used for multiple amino acid sequence alignment and phylogenetic analyses. From 2003 to 2010, CPV-2a and -2b were the only variants detected. CPV-2c, emerged in 2014, replacing CPV-2a and -2b, and has become a major variant in 2019. Phylogenetic analysis revealed that the proposed mutation pattern of VP2 amino acid residues could help distinguish Thai CPV-2 variants. This comprehensive examination provides insight into the genomic evolution of CPV-2 in Thailand since its first reporting in 2003, which may facilitate the surveillance of the potential genetic alteration of emergent CPV-2 variants.

## 1. Introduction

Canine parvoviral enteritis (CPE) is one of the leading causes of virus-mediated acute gastrointestinal illness, spreading through direct dog-to-dog contact and contamination of feces in the environment [1,2,3]. Severe hemorrhagic enteritis and dehydration in young puppies (less than 6 months) can lead to at least 80% morbidity, especially during the first five days, without proper supportive treatments [4]. Canine parvovirus type 2 (CPV-2) is the causative pathogen of CPE, belonging to the species *Carnivore protoparvovirus 1*, within the genus *Protoparvovirus* [5]. Because CPV-2 is believed to have evolved from feline panleukopenia virus (FPV), the two viruses commonly share nucleotide sequences and viral pathogenesis [6,7,8]. CPV-2 is a non-enveloped, single-stranded negative sense DNA virus [5]. Its genome consists of two open reading frames (ORFs) including ORF 1 and 2, which encode nonstructural (NS1 and NS2) and viral capsid proteins (VP1 and VP2), respectively [9].

Like other single-stranded DNA viruses, CPV-2 exhibits high genomic substitution (approximately 10^−4^ per site per year) at a rate comparable to RNA viruses [10]. Natural selective pressure between host and virus enables CPV-2 to undergo multiple genomic evolutionary events as well as the generation of novel viral variants [11]. Mutation of capsid proteins play a pivotal role in immune evasion of polyclonal neutralizing antibodies [12,13]. Because VP2 is the major capsid protein that makes up almost 90% of the entire capsid, genetic alternations in the VP2 gene have become the major focus of most evolutionary studies [14].

Within a few years following its first identification in 1977, CPV-2 underwent an antigenic shift, giving rise to the CPV-2a variant with substitutional mutations of several VP2 amino acid residues, including M87L, G300A, Y305N, and V555I [6,15,16]. An additional substitution mutation at VP2-426 amino acid residue to aspartate (D) and glutamate (E) engendered the CPV-2b and CPV-2c variants, respectively [6,17]. Additionally, new CPV-2a and -2b variants have been identified by alteration of the VP2 amino acid sequence from serine (S) to alanine (A) at amino acid residue 297, localized in the proximity of epitope B, which is speculated to be involved in the antigenicity of CPV-2 variants [17,18]. 

The temporal and geographical distribution of CPV-2 on a global scale has been constantly changing due to evolutionary fitness between host and virus [11]. Thus far, the original CPV-2 was largely replaced by three major variants: CPV-2a, -2b, and -2c [11,19,20]. In North and South America, except in Colombia and Canada, CPV-2c is the most common variant. Most countries in Asia, Europe, and Oceania have the CPV-2a variant dominantly distributed, except Japan, Australia, and New Zealand where the CPV-2b variant is prevailing [21,22,23,24,25,26,27,28,29,30].

In the past two decades, the distribution and genetic evolution of CPV-2 variants in Thailand have been reported in three separate studies in which the data were independently analyzed along different timelines, thereby lacking a comprehensive view of the evolutionary changes of CPV-2 variants [31,32,33]. In this study, we aim to provide a broad perspective of the genomic evolution and distribution of three CPV-2 variants in Thailand from its first reporting in 2003 to 2019.

## 2. Materials and Methods

### 2.1. Specimen Collection

Between February 2018 and March 2019, a total of fifteen rectal swabs were obtained from veterinary clinics in Bangkok, Thailand. Informed consent was obtained from dog owners participating in this study. All measures were taken to minimize the patient’s discomfort during sample collection. The patient’s age, date of collection, clinical presentation, and vaccine history were recorded (Appendix A). An initial screening for CPV-2 was performed using a chromatographic immunoassay, the Rapid CPV/CCV Ag test kit (AniGen, Seoul, Republic of Korea). The samples that tested positive for CPV-2 were preserved at −20 °C for further molecular biological analysis.

### 2.2. CPV-2 Genomic DNA Extraction, VP2 Gene Amplification, Sanger Sequencing and Analysis

Total DNA was extracted from CPV-2 positive specimens using Geneaid™ DNA Isolation Kit (Geneaid, New Taipei City, Taiwan) according to the manufacturer’s instructions. The full length of the VP2 gene was amplified by polymerase chain reaction (PCR) with the VP2-specific primers (cVP2-F and cVP2-R) previously described [34]. Each 25 µL PCR reaction contained 10 μL of total extracted DNA (500 ng), 0.5 μL of each forward and reverse VP2-specific primers (10 pmol), 2 μL of 10× Taq polymerase buffer, 0.5 μL of dNTPs (10 mM), 0.75 of MgCl_2_ (25 mM), and of 0.1 μL of Platinum II^®^ Taq DNA polymerase (2.5 units) (Invitrogen, Carlsbad, CA, USA). Thermal profile consisted of an initial denaturation at 98 °C for 30 s, 40 cycles of denaturation (98 °C for 10 s), annealing (59 °C for 30 s), and elongation (72 °C for 30 s), and the final cycle at 72 °C for 10 min. PCR reactions with phosphate-buffered saline (PBS) and Bayovac^®^ vaccine (Bayer Animal Health, Leverkusen, Germany) were used as negative and positive controls, respectively. The PCR products and controls were electrophoresed on a 1% agarose gel and visualized under ultraviolet light. The expected 2000 bp PCR products were excised from the agarose gel and purified using the HiYield™ Gel/PCR Fragments Extraction Kit (RBC Biosciences, New Taipei City, Taiwan). Sanger sequencing of purified PCR products was conducted at a commercial laboratory (Macrogen Inc., Seoul, Republic of Korea). The nucleotide sequences of the VP2 gene were analyzed by the Lasergene Molecular Biology software (DNA star, Madison, WI, USA) and submitted to the GenBank database.

### 2.3. CPV-2 Sequence Retrieval

Thai CPV-2 and representative nucleotide sequences were retrieved from the GenBank database. Two hundred and sixty-eight Thai CPV-2 sequences were used for classifying three CPV-2 antigenic variants based on the substitution mutation of VP2-426 amino acid residue. One hundred and twenty Thai CPV-2 and one hundred and three representative nucleotide sequences containing the full-length VP2 gene were used for multiple amino acid alignment and phylogenetic analysis. The representative sequences consisted of feline panleukopenia virus (FPV), commercial CPV-2 vaccines, and CPV-2 nucleotide sequences. FPV sequence was used as an outgroup. Only commercial CPV-2 vaccines used in Thailand were included. The representative CPV-2 sequences were derived from four continents (Asia, Europe, North America, and South America). The selection criteria for all representative sequences are indicated in Appendix A.

### 2.4. Data Analysis and Phylogenetic Tree Construction

Multiple sequence alignments were conducted by the ClustalW algorithm with the Lasergene Molecular Biology software (DNASTAR, Madison, WI, USA). Phylogenetic analysis was performed in MEGA X software (available at www.megasoftware.net) using the maximum likelihood method and Jones-Taylor-Thornton (JTT) substitution model with 1000 bootstrap replicates. Initial tree(s) for the heuristic were built by applying the nearest neighbor interchange algorithm (NNI) to a matrix of pairwise distances estimated using the maximum composite likelihood value. Phylogenetic trees were visualized using Interactive Tree of Life (iTOL), a web-based tool that can be accessed through https://itol.embl.de. Graphs and charts were generated with GraphPad and Microsoft Excel. 

## 3. Results

### 3.1. Dynamic changes of CPV-2 in Thailand between 2003 and 2019 

Since it was first reported in 2003, the CPV-2 viral genome circulating in Thailand has been dynamically changing during the past sixteen years [10]. Although CPV-2 prevalence has been reported by several groups, a comprehensive overview of CPV-2 distribution and its genomic mutations in Thailand has not yet been fully described [31,32,33]. To gain a better picture of the CPV-2 variant distribution in Thailand, two hundred and sixty-eight CPV-2 sequences were retrieved from the GenBank from 2003 to 2019 for DNA sequence analysis (Figure 1a). Based on the mutations of VP2 amino acid residue 426, the variants of CPV-2a, -2b, and -2c can be distinguished by their asparagine (N), aspartate (D), and glutamate (E) amino acid residues, respectively.

CPV-2a and -2b were the first two variants reported in 2003. From 2004 to 2010, the prevalence of CPV-2a ranged from 48% to 100%, while the distribution of the CPV-2b variant largely fluctuated between 13% and 52% (Figure 1b, Appendix A). After 2010, the level of the CPV-2b variant markedly declined to less than 4% and disappeared in 2018. CPV-2c, which first appeared in 2014, slowly prevailed over CPV-2a and -2b, and became the only variant detected in 2019. Of all samples collected from 2003 to 2019, CPV-2a made up the largest proportion of all CPV-2 variants at 46% (123/268), followed by the prevalence of CPV-2b and CPV-2c variants at 39% (104/268) and 15% (41/268), respectively (Figure 1c).

### 3.2. Genomic Evolution of CPV-2a Variant

Fifty-three Thai CPV-2a nucleotide sequences obtained from the GenBank between 2003 and 2018 were identified as the new CPV-2a variant, which was discriminated from the original CPV-2a by a serine (S) to alanine (A) substitution mutation in the VP2 capsid protein at amnio acid residue 297 [17,18]. Multiple amino acid alignments of the Thai CPV-2a sequences demonstrated five unique mutation patterns based on the degree of mutation of four VP2 amino acid residues 80, 267, 324, and 440 (Figure 2a).

The temporal distribution of CPV-2a genomic patterns has constantly changed in the past decade. The non-mutated CPV-2a (80R, 267F, 324Y, and T440T; yellow) was the only sublineage circulating in Thailand between 2003 and 2004 (Figure 2b); however, its prevalence sharply dropped from 100% (7/7) to 25% (1/3) in 2008 and disappeared in 2009. The CPV-2a with a three-mutation pattern (F267Y, Y324I, and T440A; blue), first emerged in 2008, has become the main sublineage with a prevalence ranging from 75% to 100% between 2008 and 2018. Although CPV-2a with a single mutation pattern (Y324I; green), a two-mutation pattern (Y324I and T440A; gray) and a four-mutation pattern (R80T, F267Y, Y324I, and T440A; purple) were still detected in 2010, the proportion was relatively low at 4%, 2%, and 2%, respectively (Appendix A).

To further illustrate the association between the five mutation patterns of CPV-2a and their genetic evolution, fifty-three Thai CPV-2a and forty representative VP2 amino acid sequences were used for phylogenetic analysis in which three distinctive clades were identified (Figure 2c). Clade I (blue dot) consisted of FPV, CPV-2 vaccine strains, and the original CPV-2a representative sequences with no mutation at VP2 amino acid residues 80, 267, 324, and 440. Thai CPV-2a and representative sequences with no mutation or a single-mutation pattern were gathered in clade II (pink dot), while clade III (gray dot) consisted of sequences with at least two amino acid mutations. Noticeably, CPV2a/Thailand/KP715672/2010, a sequence with a four-mutation pattern, revealed the longest branch length (0.18) within clade III (gray dot), indicating more genetic alteration than other CPV-2a sequences. 

Additionally, we observed that three representative sequences with a single mutation (CPV2a/Italy/FJ005254/2005, CPV2a/297A/South Korea/EU009200/2006, and CPV2a/South Korea/FJ197825/2007; black texts) were categorized in clade III (gray dot) as opposed to clade I (blue dot), where all sequences with a single or no mutation were presented. It appeared that these sequences contained an alanine substitution mutation of VP2-440 amino acid residue, which was different from the single mutation pattern commonly found in Thai CPV-2a carrying the tyrosine substitution mutation of VP2-324 amino acid residue (Appendix A). This finding implied that the mutation site and the type of amino acid substitution may contribute to clade classification in addition to mutation patterns.

Thai CPV-2a with a three-amino acid mutation pattern, which became a dominant sublineage from 2008 to 2018, showed a minimal amino acid change, determined by the absence of a distinctive internal node and short branch length within clade III (Figure 2c). The extended analysis of multiple nucleotide sequence alignments revealed that five Thai CPV-2a sequences isolated in 2018 contained substantial nucleotide alteration throughout the VP2 gene. In particular, the mutation with more than two nucleotides were detected in three amino acid residues: 80, 324, and 440 (Appendix A). In agreement with the multiple sequence analysis, the phylogenetic tree also displayed a remarkable node with the longest branch length (0.32) consisting of five branches of 2018 isolates (blue highlights), which was further distant from other CPV-2a sequences within clade III (Figure 2d). The result reveals that a significant nucleotide mutation at various VP2 amino acid residues was observed in recent CPV-2a isolates, while VP2 amino acid alteration remained largely unaffected.

### 3.3. Genetic Alteration of CPV-2b Variant

All Thai CPV-2b variants circulating from 2003 to 2010 were characterized as a new CPV-2b, indicated by the alanine substitution mutation of VP2-297 amino acid residue. The unique mutation patterns were noted in Thai CPV-2b sequences based on differences in the quantity and site of substitution mutation of VP2-267, VP2-324, and VP2-440 amino acid residues, which were also identified in Thai CPV-2a sequences (Figure 3a). The sequences with no mutation (267F, 324Y, and 440T; yellow) were the only one sublineage observed in Thai CPV-2b variants from 2003 to 2008 but were no longer detected in 2010. (Figure 3b). The emergence of CPV-2b with a two-mutation pattern (F267Y andY324I; gray) became the dominant sublineage with 87% (27/31) prevalence in 2010 and made up 71% (27/38) of all CPV-2b mutation patterns reported in Thailand from 2003 to 2010. (Figure 3b, Appendix A). Thai CPV-2b sequences from 2016 to 2017 were excluded from data analysis, as the sequences lacked information at VP2 amino acid residues 267 and 324, which were essential for mutation pattern analysis. 

Thirty-eight Thai CPV-2b and thirty-four representative VP2 amino acid sequences were used for phylogenetic analysis in which five prominent clades were noted. Clade I (blue dot) contained FPV, CPV-2 vaccine strains, and the original CPV-2b sequences without the substitution mutation of VP2-267, VP2-324, and VP2-440 amino acid residues. Clades II to V were clustered based on the CPV-2b mutation patterns in which the individual clade contained the sequences sharing the same mutation pattern. 

CPV-2b/Vietnam/AB054221/1997 and CPV-2b/Chile/MT585713/2019 (black texts), exhibited the substitution mutation of VP2-440 and VP2-324 amino acid residues, respectively, which were distinct from a single mutation pattern carrying the mutation of VP2-267 residue (Appendix A). Although these two sequences had the same degree of amino acid mutation as a single mutation pattern, they were clustered in clade II (pink dot) in preference to clade III (green dot) where the CPV-2b sequences containing a single mutation were classified.

### 3.4. The Emergence of CPV-2c Antigenic Variant

The prevalence of CPV-2c variant has continued to rise since 2014 and eventually replaced the CPV-2a and -2b variants in 2019 (Figure 1b). Consistently, a rise in the CPV-2c variant has also been reported in North America, Europe, and Asia [35,36,37,38]. Several studies have further classified the CPV-2c variant into two groups based on differences in three distinct VP2 amino acid residues. A European CPV-2c consisted of 267F, 324Y, and 370G, while an Asian CPV-2c contained 267Y, 324I, and 370R [39]. Between 2014 and 2019, all CPV-2c variants detected in Thailand were Asian CPV-2c, which is widely distributed throughout Asia, including Vietnam, China, and Taiwan [22,35,39]. 

Multiple amino acid alignment of twenty-nine Thai CPV-2c sequences based on the alanine substitution mutation of VP2-5 and the isoleucine substitution mutation of VP2-447 amino acid residues were used to categorize Thai CPV-2c into three mutation patterns (Figure 4a). Thai CPV-2c with a single mutation (A5G; green) was the first pattern identified in two specimens in 2016, but its prevalence sharply dropped from 100% (2/2) to 4% (1/19) and to 20% (1/4) in 2018 and 2019, respectively (Figure 4b, Appendix A). A pattern with no mutation (5A and 447I; yellow) was first detected in 2018 at 86% (19/22). Despite the complete disappearance in 2019, the non-mutated sublineage made up the largest proportion of all CPV-2c variants reported in Thailand from 2016 to 2019 at 65% (19/29). Most recently, Thai CPV-2c isolated in 2019 demonstrated the highest prevalence of two mutation pattern (A5G and I447M; gray) at 80% (4/5) (Figure 4b, Appendix A). 

The phylogenetic analysis of CPV-2c variant revealed three distinct clades. Clade I (gray dot) comprised of FPV, CPV-2 vaccine strains, and CPV-2 sequences with no mutation of VP2-5, VP2-297, VP2-300, VP2-305 and VP2-447 amino acid residues. Asian and European CPV-2c variants were classified into Asian (pink dot) and European clades (blue dot), respectively (Figure 4c, Appendix A). All European representative sequences showed no mutation of VP2-5 and VP2-447 amino acid residues (yellow texts) and lacked internal nodes within the clade. Three internal nodes within the Asian clade were markedly separated in correspondence with the unique three mutation patterns. In each designated internal node, Thai CPV-2c and representative sequences from various countries were clustered based on the mutation patterns regardless of the geographical origin of CPV-2c sequences.

## 4. Discussion

Our study is the first to provide a comprehensive overview of the genetic evolution of CPV-2 in Thailand since its first reporting in 2003 [31]. Like RNA viruses, constant substitution mutation of CPV-2 facilitates the natural selection of certain variant(s) circulating in dog populations [40]. The CPV-2 genetic mutation and distribution landscape has continued to evolve in the past two decades. The distribution of CPV-2a and CPV-2b has been gradually replaced by the CPV-2c variant, which has finally become the most frequently observed variant of CPV-2 in Thailand in 2019. Similarly, the emerging pattern of the CPV-2c variant was also reported in neighboring countries in the Southeast Asia region such as Vietnam [22] and Laos [35] as well as other parts of the world including North America [36] and South America [29].

Most genetic studies on CPV-2 have mainly focused on the VP2 non-synonymous substitution mutation, which is known to play a role in antigenic shift and immune evasion. In this study, we observed changes in CPV-2a nucleotide sequences recently isolated in 2018, particularly in the amino acid residues associated with transferrin receptor engagement such as VP2-80, VP2-267, VP2-324, and VP2-440 [41]. By monitoring nucleotide mutations, it may be possible to predict the emergence of novel CPV-2 sublineages before amino acid changes occur.

CPV-2b with no substitution mutation of VP2-267, VP2-324, and VP2-440 amino acid residues were the only one sublineage spreading in Thailand from 2003 to 2008. Two years later, the Thai CPV-2b mutation pattern simultaneously switched from a non-mutation to a two-mutation pattern, which contained the substitution mutation of VP2-267 and VP2-324 amino acid residues, situated in a high antigenic motif essential for viral entry via transferrin receptor binding region [11,41]. Although there is no direct evidence to support the impact of VP2-267 and VP2-324 amino acid residues on the phenotypic switch from a non-mutation to a two-mutation pattern, a recent selection pressure analysis of CPV-2 revealed the association between these two amino acid residues in CPV-2b positive selection, which may favor host immune evasion [42].

Most recent CPV-2 isolated in 2019 revealed the increasing prevalence of CPV-2c variant in Thailand. Similarly, a recent systematic review also showed a rise in CPV-2c in Asia, South America, North America, and Africa [19]. Alanine substitution mutation at VP2-5 and glutamine substitution mutation at VP2-370 amino acid residues were frequently detected in the CPV-2c variant reported in several parts of Asia [22,35,39]. Only a few sequences of CPV-2c containing VP2-5 and VP2-370 residues have been detected in Europe (Italy and Romania) and Africa (Nigeria and Egypt) [19,37,43,44,45]. In addition to the mutation of VP2-5 and VP2-370, Thai CPV-2c containing isoleucine substitution mutation at VP2-447 amino acid residue has continued to increase in the past few years. Coincidently, a rise in CPV-2c variant with mutated VP2-447 was also reported in Vietnam, a Southeast Asian country close to Thailand [22,23]. We speculate that the CPV-2c variant with mutated VP2-447 might spread locally between Thailand and Vietnam through dog importation. 

In this study, the mutation patterns were proposed based on the degree and type of amino acid substitution mutations detected in Thai CPV-2 variants that circulated between 2003 and 2019. It should be noted that some of these mutation patterns were also observed in CPV-2 variants from other geographical locations, suggesting that these patterns are not exclusively characteristic of Thai CPV-2. Despite the cluster of sequences with identical mutation patterns within the same clades, we cannot interpret this finding as an evolutionary change. This is due to the limited number and geographical origin of the sequences used to perform the analysis.

Because almost all VP2 sequencing data were obtained from dogs in the Bangkok metropolitan area, it remains inconclusive whether our data fully represents the nationwide distribution of CPV-2 variants. An in-depth study with samples collected from different regions of Thailand would be pivotal in addressing several unanswered questions regarding geographical distribution; (1) Is CPV-2c still a major variant in Thailand? (2) Did CPV-2a and -2b completely vanish from Thailand? (3) Are there any novel substitution mutation patterns of CPV-2a, -2b, and -2c in other parts of Thailand, especially in the areas adjacent to Thailand’s neighboring countries? 

## 5. Conclusions

The interplay between host and virus engenders the selective pressure that allows certain CPV-2 variants to persist and disseminate among dog populations. This study portrays the genetic changes and temporal distribution that have occurred in CPV-2 variants in Thailand over the past sixteen years. Since 2019, CPV-2c has replaced CPV-2a and CPV-2b, first reported in 2003. Our retrospective outlook on the distribution of CPV-2 variants in Thailand presents in-depth information that might be useful for monitoring novel mutation patterns and emerging variants of CPV-2 in Thailand in the future.

## Figures and Tables

**Figure 1 pathogens-11-01460-f001:**
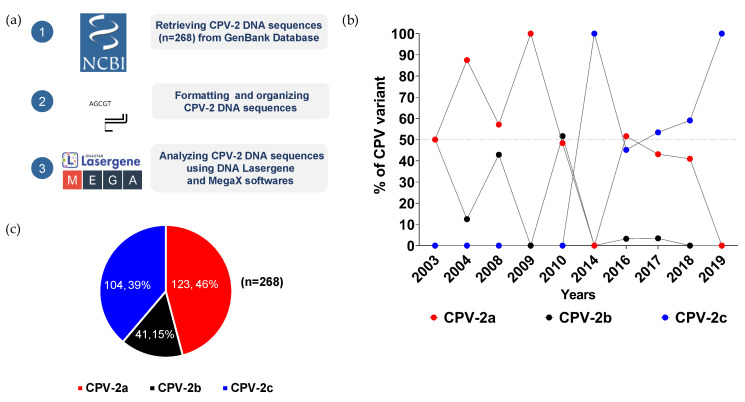
**Distribution of CPV-2 variants in Thailand.** (**a**) A schematic diagram described the workflow of sample acquisition and analysis. (**b**) Temporal distributions of CPV-2a (red dot), CPV-2b (black dot), and CPV-2c (blue dot) variants from 2003 to 2019 are illustrated in a line chart in which years and percentages are shown on the X- and Y-axis, respectively. (**c**) Overall proportions of CPV-2a (red), CPV-2b (black), and CPV-2c (blue) are depicted in a pie chart by numbers and percentages.

**Figure 2 pathogens-11-01460-f002:**
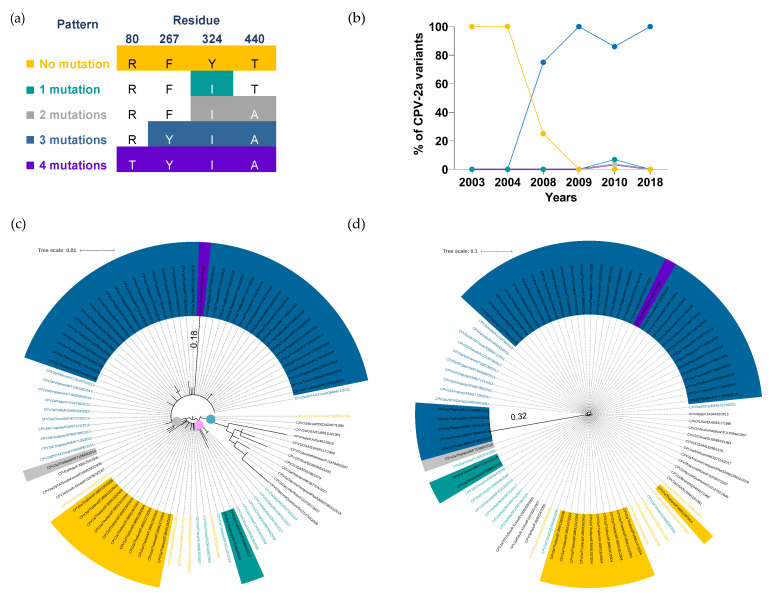
**Temporal distribution and phylogenetic analysis of CPV-2a variant.** (**a**) Four VP2 amino acid residues were used for characterizing CPV-2a into five unique patterns, including (1) no mutation (80R,267F,324Y, and T440T; yellow), (2) single mutation (Y324I; green), (3) two mutations (Y324I and T440A; gray), (4) three mutations (F267Y, Y324I, and T440A; blue), and (5) four mutations (R80T, F267Y, Y324I, and T440A; purple). (**b**) A line chart demonstrates the temporal distributions of CPV-2a mutation patterns in which years and percentages are shown on the X- and Y-axis, respectively. (**c**) A circular phylogenetic tree depicts CPV-2a amino acid sequences clustered into clade I (blue dot), clade II (pink dot), and clade III (gray dot). (**d**) A circular cladogram illustrates CPV-2a nucleotide sequences. Tree scales (0.01 and 0.1) indicate branch length shown in the top left corner. Colored highlights and texts corresponding to five mutation patterns are shown in Thai CPV-2a and representative sequences, respectively.

**Figure 3 pathogens-11-01460-f003:**
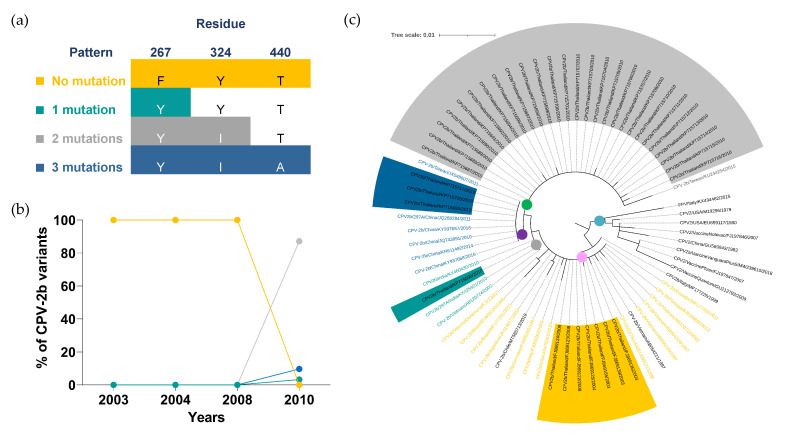
**Distribution and phylogenetic analysis of the CPV-2b variant.** (**a**) Four mutation patterns of CPV-2b were categorized based on three VP2 amino acid residues, including (1) no mutation (267F, 324Y, and 440T; yellow), (2) single mutation (Y324I; green), (3) two mutations (F267Y and Y324I; gray) and (4) three mutations (F267Y, Y324I, and T440A; blue). (**b**) Distributions of CPV-2b mutation patterns are shown in a line chart in which years and percentages are indicated on the X- and Y-axis, respectively. (**c**) A circular cladogram consists of five distinct clades. Clade I to V are displayed in blue, pink, yellow, purple, and green dots, respectively. Tree scale (0.01) indicates branch length shown in the top left corner. Colored highlights and texts corresponding to four mutation patterns are shown in the sequences derived from Thai CPV-2b and representative sequences, respectively.

**Figure 4 pathogens-11-01460-f004:**
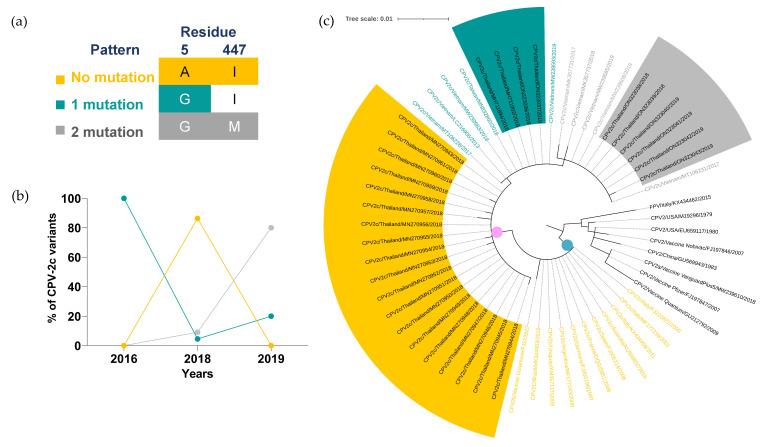
**Temporal distribution and phylogenetic analysis of CPV-2c variant.** (**a**) Three mutation patterns of CPV-2c were categorized based on three VP2 amino acid residues, including (1) no mutation (5A and 447I; yellow), (2) single mutation (A5G; green), and (3) two mutations (A5G and I447M; gray). (**b**) The percentages of each mutation pattern from 2016 to 2019 are shown in a line chart. (**c**) Clade I, European CPV-2c clade, and Asian CPV-2c clade are shown in gray, blue, and pink dots, respectively. Tree scale (0.01) indicates branch length shown in the top left corner. Colored highlights and texts corresponding to three mutation patterns are shown in the sequences derived from Thai CPV-2c and representative sequences, respectively.

## Data Availability

The sequence accession numbers used in this study are included in supplementary materials.

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
