# Peer review of "Tracing the Genetic Evolution of Canine Parvovirus Type 2 (CPV-2) in Thailand"

_pathogens, 2022, doi:10.3390/pathogens11121460_

Round 1

Reviewer 1 Report

The article entitled “Tracing genetic evolution of Canine Parvovirus Type 2 (CPV-2) 2 in Thailand” submitted to Pathogens, contributed by Tippawan Jantafong et al., describes a retrospective integrated analysis of CPV-2 genomic mutations in Thailand from 2003 to 2019. The study is interesting as it provides a comprehensive view of  the genomic evolution of CPV-2 with wie interest for the Parvoviridae field.

Overall the work has been properly conducted, the phylogenetic analysis was performed adequately, and the results support the major conclusions. The article is well written and the figures in general have been elaborated at high quality.

There are a few points though that authors may need to pay attention to improve the clarity of the message.

Comments:

  1. I face problems to open the link of the Supplementary figures.
  2. In Materials and Methods, the authors for PCR extension use elongation at 72 °C for 30 sec for an expected 2000 bp product. This seems like a short time for the Taq Platinum enzyme they use. Are these the real conditions?, or did they use the Platinum II enzyme? Please review this point.
  3. The quality and size of the images should allow to see the access number of the sequences more easily.

Author Response

November 6, 2022

Re: Manuscript ID: pathogens-1991050

Dear Reviewer,

We would like to thank you for considering the manuscript we submitted for publication.  Your valuable time and insightful comments are greatly appreciated and extremely helpful to strengthen the quality of the manuscript. The manuscript has been revised in response to your suggestions as follows.

  1. I face problems to open the link of the Supplementary figures.

It is our regret that you were unable to access the supplementary materials that were uploaded along with the manuscript. In this revised manuscript, the supplementary materials have been included after the reference section for your convenience.

  1. In Materials and Methods, the authors for PCR extension use elongation at 72 °C for 30 sec for an expected 2000 bp product. This seems like a short time for the Taq Platinum enzyme they use. Are these the real conditions?, or did they use the Platinum II enzyme? Please review this point.

The Platinum II enzyme was used for amplifying VP2 gene. A revision has been made to the materials and methods section. 

  1. The quality and size of the images should allow to see the access number of the sequences more easily.

There is little chance of adjusting the font size of the phylogenetic trees since they were produced with the largest possible font size. Despite being the fact that all figures were created and saved at 1200 dpi, the PDF file converted from docx might affect the image clarity. We will coordinate with the MDPI’s graphics team to ensure high-quality figures. 

It is our pleasure to send you this revised manuscript and we hope that all concerns and questions have been addressed adequately. To facilitate reviewer viewing, all changes have been marked up using the "Track Changes" function. We would be glad to respond to any further questions and comments that you may have. Your attention to this study is greatly appreciated, and we hope that the revised manuscript is suitable for publication.

Please do not hesitate to contact me for any further clarification.

Sincerely,

Krit Ritthipichai, DVM, PhD

Reviewer 2 Report

Journal: Pathogens

Manuscript ID: pathogens-1991050

Title: Tracing genetic evolution of Canine Parvovirus Type 2 (CPV-2) 2 in Thailand

1. Overview and general recommendation:

In this study, Authors analysed fifteen rectal swabs collected from dogs in Bangkok, Thailand. CPV-2 VP2 nucleotide sequences were obtained and these sequences were analysed along with those related from Thailand and few CPV-2 reference sequences, both obtained from the GenBank database. Authors report the sequence and phylogenetic analyses of CPV-2 in Thailand between 2003 and 2019, focusing their attention on the CPV-2 evolutionary pathways.

This study, compact but in some parts redundant and not very smooth, aimed to depict the genetic evolution of CPV-2 in Thailand but it appears limited in the discussion/conclusion parts and needs to be revised to improve the overall description and to solve some critical issues. Indeed, it lacks of scientific soundness in some parts and, mainly, of a clear discussion and consequential conclusions. 

I added some comments and suggestions to Authors to improve the description throughout the manuscript. I explained these comments and suggestions in more details below.

2.1 Major comments:

- As the analysis is biased by the limited number of reference sequences, it appears really speculative the sentence “Phylogenetic analysis revealed that the newly proposed mutation pattern of VP2 amino acid residues can be used for distinguishing sublineage of CPV-2 variants, as indicated by the formation of distinct clades.”. Indeed, it is almost obvious that changes at few single amino acid residues in a such limited dataset depict small to medium clades in the phylogenetic tree but this cannot be so easily considered as an evolutionary point of view but it only reflects these single mutations. This explains why sequences with single amino acid changes but at different residues could lies in the same or in close related clades. I suggest to carefully revise this part. 

- It is not clear the relationship between the analysis of the CPV-2 genetic evolution in Thailand and how it “helps direct the course of action for evaluating vaccine efficacy”. I suggest to carefully revise this conclusion, according to the data analysed in this study.

- Lines 32-35: CPV taxonomy, genus/species level, and the use of italics characters, when necessary, should be revised. The concept of shared “evolutionary fitness” at line 37 should be also revised because FPV showed a different evolutionary fitness over the years.

- Lines 43-45: I suggest to remove “and evolution” and to include the CPV-2 mutation rates reported in the literature.

- Lines 55-58: I suggest to revise the VP2-426 amino acid residues for the CPV-2b and CPV-2c variants, and to remove “from alanine (A)”; moreover, I suggest to replace “CPV-new2a” or “CPV-new2b” with “new CPV-2a” or “new CPV-2b”, respectively, both here and throughout the whole manuscript.

- According to the guidelines for Authors, Material and Methods section may be divided by numbered subheadings, figures “should be placed in the main text near to the first time they are cited”, Data Availability Statement should be implemented with the sequence accession numbers for the CPV strains from the analysis of this study, and the Author Contributions should be revised according to the guidelines.

- In the whole Results sections are included parts that could be considered as discussions (i.e., lines 119-123, 138-143, 160-164, etc.) as well as references, that usually are not included in this section. Inclusion or exclusion criteria for retrieving sequences from GenBank were not included (i.e., country of origin, complete VP2 gene sequences, etc.). The total number of each CPV-2 variant was not included neither in the text (lines 126-128) nor in the Figure 1. At lines 129-131, the ranges of percentages should be included instead of a generic “more than 48%”. “Major” at line 129 should be revised because it is not clear which is the “minor” variant.  The text from line 150 to 193 is difficult to follow as well as at lines (196-224 and 237-251) referred to CPV-2b and -2c variants, respectively: I suggest to remodulate the flow of the text to be easier to read for the potential audience. Conclusions at lines 182-183 and 193-195 should be placed in the proper paragraph and better explained.

- Discussions at lines 388-394 and 404-411 appears vague and not follow the evidences of this study: I suggest to carefully revise these parts. Moreover, despite reported in the literature, this study do not allow to evaluate the immune evasion by specific CPV-2 mutants and globally ignored the increasing rates of the so called Asian CPV-2c variant in comparison with the increasing rates in other Asian countries or did not consider its worldwide spread in the same years.  

2.2 Minor comments:

- Title: lowercase letters should be used for Canine, Parvovirus, and Type.

- Line 10: again, lowercase letter should be used for Parvovirus.

- Line 11: I suggest to use the past tense for “contribute”.

- Line 12: I suggest to replace “quantified” with “analysed”.

- Line 19: I suggest to add a comma after “-2b” and to remove “eventually”; moreover, since this study was conducted until 2019, “it has become a major variant as of 2019” deserves to be revised.

- Line 22: I suggest to replace “Our” with “This” and to add “in Thailand” after “CPV-2”.

- Lines 35-36: “Because CPV-2 is directly descended from FPV” deserves to be revised.

- Line 39: please, add “for” after “encode” and remove “proteins” after “nonstructural”.

- Lines 40-42: this part could be removed because is not relevant for introducing this study.

- Line 83: I suggest to remove “CPV-2 genomic”, “DNA” and “and analysis”.

- Line 86: the commercial name of the extraction kit should be added.

- Lines 87-88: I suggest to revise with “..amplified by a polymerase chain reaction (PCR) assay using the VP2-specific primers pair (specify the names of the primers; cVP2-F/cVP2-R?) previously described [38].”.

- Lines 90-91: I suggest to include the name of the amplification kit and the volumes expressed in µl other than the concentration. I suggest to replace “PCR program” with “Thermal profile”.

- Line 98: again, please include the name of the purification kit.

- Line 101: replace with “DNASTAR”

- Lines 106-107: I suggest to remove “in”, to place “Asia,..America” in brackets, and to remove “in FASTA format”.

- Line 108: “were conducted using…algorithm with the Lasergene…”

- Line 132: “eventually” could be removed.

- Line 133: “dominated” could be replaced by “prevailed over”

- Line 134: add “one” before “variant”.

- Figure 1: can Authors reproduce the NCBI and DNASTAR images? Use lowercase a,b,c letter in the caption. Replace “demonstrates” with “describes”, “samples” with “sequences”. Remove the text from “Two hundred…” to “..(E), respectively.”. Replace here and in the main text “closed circles” with “dots”. “numbers” at line 263 are not included. I suggest to change one color between blue or green (i.e., with red) to be clearer.

- Figure 2: Use lowercase a,b,c letter in the caption. Remove “Fifty-three…alignment.” and “Phylogenetic analysis…replicates.”

- Figure 3: add “the” before “CPV-2b” and change “sublineages” with “variant”. Remove “Thirty-eight…alignment” and “Phylogenetic analysis…replicates.”

- Figure 4: change “CPV-2b” with “CPV-2c” at line 362. Remove “Thirty-nine…alignment” and “Phylogenetic analysis…replicates.”

Author Response

November 6, 2022

Re: Manuscript ID: pathogens-1991050

Dear Reviewer,

We would like to thank you for considering the manuscript we submitted for publication.  Your valuable time and insightful comments are greatly appreciated and extremely helpful to strengthen the quality of the manuscript. The manuscript has been revised in response to your suggestions as follows.

2.1 Major comments:

  1. As the analysis is biased by the limited number of reference sequences, it appears really speculative the sentence “Phylogenetic analysis revealed that the newly proposed mutation pattern of VP2 amino acid residues can be used for distinguishing sublineage of CPV-2 variants, as indicated by the formation of distinct clades.”. Indeed, it is almost obvious that changes at few single amino acid residues in a such limited dataset depict small to medium clades in the phylogenetic tree but this cannot be so easily considered as an evolutionary point of view but it only reflects these single mutations. This explains why sequences with single amino acid changes but at different residues could lies in the same or in close related clades. I suggest to carefully revise this part. 

A reviewer's suggestion was incorporated into the results and discussion sections.

  1. It is not clear the relationship between the analysis of the CPV-2 genetic evolution in Thailand and how it “helps direct the course of action for evaluating vaccine efficacy”. I suggest to carefully revise this conclusion, according to the data analysed in this study.

In response to the reviewer's recommendations, the conclusion have been revised.

  1. Lines 32-35: CPV taxonomy, genus/species level, and the use of italics characters, when necessary, should be revised.

Revisions have been made based on the suggestions from the reviewer.

  1. The concept of shared “evolutionary fitness” at line 37 should be also revised because FPV showed a different evolutionary fitness over the years.

The concept of evolutionary fitness has been revised.

  1. Lines 43-45: I suggest to remove “and evolution” and to include the CPV-2 mutation rates reported in the literature.

Line 43-45 has been revised per reviewer’s suggestion.

  1. Lines 55-58: I suggest to revise the VP2-426 amino acid residues for the CPV-2b and CPV-2c variants, and to remove “from alanine (A)”

Revised to “An additional substitution mutation of VP2-426 amino acid residue to asparagine (N) engendered the CPV-2b variant”.

  1. Moreover, I suggest to replace “CPV-new2a” or “CPV-new2b” with “new CPV-2a” or “new CPV-2b”, respectively, both here and throughout the whole manuscript.

new CPV-2a” and “new CPV-2b were replaced throughout the whole manuscript.

  1. According to the guidelines for Authors, Material and Methods section may be divided by numbered subheadings

Subheadings have been numbered in both material and method sections. 

  1. Figures “should be placed in the main text near to the first time they are cited”, 

The figures have been reorganized in the main text near to the first time they are cited. 

  1. Data Availability Statement should be implemented with the sequence accession numbers for the CPV strains from the analysis of this study,

The statement has been included in the data availability section.

  1. the Author Contributions should be revised according to the guidelines.

The author contributions have been revised per the journal guidelines.

  1. In the whole Results sections are included parts that could be considered as discussions (i.e., lines 119-123, 138-143, 160-164, etc.) as well as references, that usually are not included in this section.

In response to the reviewer's recommendations, the conclusion and discussion sections have been revised.

  1. Inclusion or exclusion criteria for retrieving sequences from GenBank were not included (i.e., country of origin, complete VP2 gene sequences, etc.).

Information was included in Materials and Methods section and Supplementary Table 2.  

  1. The total number of each CPV-2 variant was not included neither in the text (lines 126-128) nor in the Figure 1.

The total number of each CPV-2 variant has been incorporated throughout the main text as well as supplementary Tables.

  1. At lines 129-131, the ranges of percentages should be included instead of a generic “more than 48%”.

The ranges of percentages have been added.

  1. “Major” at line 129 should be revised because it is not clear which is the “minor” variant.

 “Major” has been removed.     

  1. The text from line 150 to 193 is difficult to follow as well as at lines (196-224 and 237-251) referred to CPV-2b and -2c variants, respectively: I suggest to remodulate the flow of the text to be easier to read for the potential audience.

The flow of result section has been reorganized.  

  1. Conclusions at lines 182-183 and 193-195 should be placed in the proper paragraph and better explained.

Revisions have been made based on the suggestions from the reviewer.

  1. Discussions at lines 388-394 and 404-411 appears vague and not follow the evidences of this study: I suggest to carefully revise these parts. Moreover, despite reported in the literature, this study do not allow to evaluate the immune evasion by specific CPV-2 mutants and globally ignored the increasing rates of the so called Asian CPV-2c variant in comparison with the increasing rates in other Asian countries or did not consider its worldwide spread in the same years.  

The discussion section has been revised considering the reviewer's suggestions.

Minor comments:

-Title: lowercase letters should be used for Canine, Parvovirus, and Type.

Changed to “Tracing genetic evolution of canine parvovirus type 2 (CPV-2) in Thailand”.

- Line 10: again, lowercase letter should be used for Parvovirus.

Changed to “Canine parvovirus type 2”.

- Line 11: I suggest to use the past tense for “contribute”.

Changed to “High genomic substitution rates in CPV-2 contributed to”.

- Line 12: I suggest to replace “quantified” with “analysed”.

Replaced with analyzed.

- Line 19: I suggest to add a comma after “-2b” and to remove “eventually”; moreover, since this study was conducted until 2019, “it has become a major variant as of 2019” deserves to be revised.

Changed to “CPV-2c, emerged in 2014, replaced CPV-2a and -2b, and has become a major variant in 2019”.

- Line 22: I suggest to replace “Our” with “This” and to add “in Thailand” after “CPV-2”.

Replaced with This and added Thailand.

- Lines 35-36: “Because CPV-2 is directly descended from FPV” deserves to be revised.

Changed to “Because CPV-2 is believed to have evolved from FPV, the two viruses commonly share nucleotide sequences and viral pathogenesis”.

- Line 39: please, add “for” after “encode” and remove “proteins” after “nonstructural”.

Added for “for” after “encode” and removed “proteins” after “nonstructural”.

- Lines 40-42: this part could be removed because is not relevant for introducing this study.

Lines 40 to 42 were removed.

- Line 83: I suggest to remove “CPV-2 genomic”, “DNA” and “and analysis”.

Removed “CPV-2 genomic”, “DNA” and “and analysis”.

- Line 86: the commercial name of the extraction kit should be added.

The commercial name of the kit was added.

- Lines 87-88: I suggest to revise with “..amplified by a polymerase chain reaction (PCR) assay using the VP2-specific primers pair (specify the names of the primers; cVP2-F/cVP2-R?) previously described [38].”.

Primer names were added.

- Lines 90-91: I suggest to include the name of the amplification kit and the volumes expressed in µl other than the concentration. I suggest to replace “PCR program” with “Thermal profile”.

Included the name of the amplification kit and the volume and replaced “PCR program” with “Thermal profile”.

- Line 98: again, please include the name of the purification kit.

The name of the purification kit was included.

- Line 101: replace with “DNASTAR”

Replaced with DNASTAR.

- Lines 106-107: I suggest to remove “in”, to place “Asia,..America” in brackets, and to remove “in FASTA format”.

Removed “in”, to place “Asia,..America” in brackets, and to remove “in FASTA format”.

- Line 108: “were conducted using…algorithm with the Lasergene…”

Changed to “CPV-2a, CPV-2b, and CPV-2c from other countries in (Asia, Europe, North America, and South America) were retrieved from the GenBank database”.

- Line 132: “eventually” could be removed.

Removed

- Line 133: “dominated” could be replaced by “prevailed over”

Replaced with “prevailed over”

- Line 134: add “one” before “variant”.

Added “one”

Figure 1:

-Can Authors reproduce the NCBI and DNASTAR images?

NCBI and DNASTAR images were reproduced

-Use lowercase a,b,c letter in the caption.

Used lowercase letters in the caption

-Replace “demonstrates” with “describes”, “samples” with “sequences”.

Replaced “demonstrates” with “describes”, “samples” with “sequences”.

-Remove the text from “Two hundred…” to “..(E), respectively.”.

Removed the text.

-Replace here and in the main text “closed circles” with “dots”. “numbers” at line 263 are not included.

Replaced “closed circles” with dots in the main text and figure legends.

-I suggest to change one color between blue or green (i.e., with red) to be clearer.

Changed the colors

Figure 2:

-Use lowercase a,b,c letter in the caption. Remove “Fifty-three…alignment.” and “Phylogenetic analysis…replicates.”

Used lowercase a,b,c letter in the caption and removed “Fifty-three…alignment.” and “Phylogenetic analysis…replicates.”

Figure 3:

-Add “the” before “CPV-2b” and change “sublineages” with “variant”. Remove “Thirty-eight…alignment” and “Phylogenetic analysis…replicates.”

Added “the” before “CPV-2b” and change “sublineages” with “variant” and removed “Thirty-eight…alignment” and “Phylogenetic analysis…replicates.”

Figure 4:

-Change “CPV-2b” with “CPV-2c” at line 362. Remove “Thirty-nine…alignment” and “Phylogenetic analysis…replicates.”

Changed “CPV-2b” with “CPV-2c” at line 362 and removed “Thirty-nine…alignment” and “Phylogenetic analysis…replicates.”

It is our pleasure to send you this revised manuscript and we hope that all concerns and questions have been addressed adequately. To facilitate reviewer viewing, all changes have been marked up using the "Track Changes" function. We would be glad to respond to any further questions and comments that you may have. Your attention to this study is greatly appreciated, and we hope that the revised manuscript is suitable for publication.

Please do not hesitate to contact me for any further clarification.

Sincerely,

Krit Ritthipichai, DVM, PhD

Round 2

Reviewer 2 Report

Journal: Pathogens

Manuscript ID: pathogens-1991050

Title: Tracing genetic evolution of canine parvovirus type 2 (CPV-2) 2 in Thailand

1. Overview and general recommendation:

I revised this manuscript for this second review stage, and I found it little improved but still some criticisms should be addressed. I added some comments and suggestions to Authors to improve the description throughout the manuscript. I explained these comments and suggestions in more details below.

2.1 Major comments:

- Authors need to consider that observed amino acid changes are not exclusively the result of evolutionary driving forces necessarily restricted to Thailand but could be also due to the sudden introduction from other not analysed or distant areas/countries. Therefore, I suggest to not exclusively describe this study, also considering the limited number of the included sequences, only as the description of a genetic evolution.

- As in the previous comment, due to the type of this study, speculations on the immune escape or any limits of current use vaccines are unnecessary (as at lines 425-432 and 483-484).

- Lines 33-34: Authors should revise this part referred to “genogroups” and CPV-1, considered as FPV. This is very important because this taxonomical designation is not correct.  

- Line 53: As in the previous review stage, Authors should revise the amino acid residue for the CPV-2c variant.

- Line 54: I suggest to replace with “Additionally, ..”, because these two genetic variants have been observed less recently than the CPV-2c variant.

- Lines 126-134: this part (from “due to a very..”) not includes results and, therefore, could be removed.

- Lines 144-149: similarly, this part should be included in the discussion section, not in this section.

- Lines 194-197 and 323-325: these are obvious considerations rather than informative conclusions and, therefore, they could be removed.

- Line 457: The meaning of “the prevalence was negligible” is not clear; moreover, references of these studies are missing.

2.2 Minor comments:

- Line 12: I suggest to add “previous” before “studies”

- Line 15: I suggest to replace “our” with “this”

- Line 21: I suggest to remove “into different sublineages”

- Line 40: I suggest to replace with “Despite a DNA virus,..”

- Line 62: replace with “..have the CPV-2a variant dominantly..”

- Line 63: replace with “..where the CPV-2b variant is..”

- Lines 83-84: replace with “…by a polymerase chain reaction (PCR) assay using VP2-specific primers (cVP2-F and cVP2-R) previously…”

- Line 101: replace with “Thai and reference CPV-2 nucleotide..”

- Lines 103-104: “based on…amino acid residue.” could be removed. “representative” could be replaced by “reference” (also at lines 106, 109, and 186)

- Line 105: replace with “..sequences, were used…”

- Line 106: replace with “..sequences included feline …”

- Line 137: “After” could be replaced by “Since”

- Line 139: “its predecessors,” could be removed.

- Figure 1: I suggest to remove the subfigure “a” because not more informative than the main text.

- Line 175: I suggest to replace “mutation” with “genomic”

- Line 433: I suggest to replace with “..genetic studies on CPV-2..”

- Line 434: I suggest to use “mutations”

- Line 435: I suggest to remove “considerable”

- Line 438: “has not yet…acid residues” is not exact and not clear.

- Lines 441 and 444: please revise “substitution mutation”

- Lines 453-454 and 458: I suggest to replace “substitution mutation of” with “at”

- Line 480: I suggest to remove “evolution” and to use “changes”

- Line 482: I suggest to replace “and” with “or” and to add “could” before “determine”

Author Response

Dear Reviewer,

The manuscript has been revised in response to your suggestions. Please find the attached file. 

Round 3

Reviewer 2 Report

Authors have improved the overall quality of the manuscript, by including most of the suggestions. At this stage of the revision, no other suggestions are necessary by the Reviewer.    

Author Response

Thank you very much for your valuable time and insightful comments.